# A Resistivity Plate Loading Device for Assessing the Factors Affecting the Stiffness of a Cement-Stabilized Subgrade

**DOI:** 10.3390/ma15103453

**Published:** 2022-05-11

**Authors:** Huaiping Feng, Ackah Frank Siaw, Hailiang Wang

**Affiliations:** 1State Key Laboratory of Mechanical Behavior and System Safety of Traffic Engineering Structures, Shijiazhuang Tiedao University, Shijiazhuang 050043, China; fenghuaiping@stdu.edu.cn (H.F.); wanghailiang@gmail.com (H.W.); 2Key Laboratory of Mechanical Behavior Evolution and Control of Traffic Engineering Structures in Hebei, Shijiazhuang Tiedao University, Shijiazhuang 050043, China

**Keywords:** resistivity plate loading test, water content, cement hydration, subgrade reaction modulus

## Abstract

The extent of mixing in the stabilization process and the control of the cement content (*C*) and water content (*w*) in the mixture are key to the outcome of the engineering performance of a cement-stabilized subgrade. Intelligent Compaction (IC) quality control has improved quality control and management practices during construction. Intelligent Compaction Measurement Values (ICMVs) selected to evaluate the stiffness properties of cement-stabilized soils do not directly relate to the stiffness properties of the cement-stabilized subgrade and do not consider *w* and *C*. Additional tests need to be conducted for calibration of ICMVs. In this study, our solution is the development of a resistivity plate loading test. The resistivity plate loading test features the flexibility in determining the soil stiffness, *w*, *C,* and other important factors, such as the time of test effect (hydration) (*T*) and dry density (*ρ*_d_). To verify the accuracy of the testing method, laboratory experimental studies were conducted on cemented soils considering *ρ*_d_, *w*, *C,* and *T* at different factor levels. Multiple response studies based on grey rational analysis (GRA) were conducted. Analysis of the input factors was performed, and their effects on the measured responses were quantified. According to the study, the *ρ* measured by the device was a powerful indicator of stiffness, *ρ*_d_, *w*, *C,* and *T**,* which showed that the device can be useful equipment for quality control and an advancement in the in situ testing technologies and test equipment. A statistical regression model based on the linear and linear plus interaction terms among the factors is proposed to predict the average responses.

## 1. Introduction

Soil stabilization with cement is a well-established practice for constructing rail/roadway embankments and pavement layers. Cement stabilization allows the improvement of both standard soils and substandard in situ soils to levels consistent with the construction requirements. The stiffness and performance of stabilized subgrades are related to how close the stabilized subgrade is to achieving optimal compaction properties. As a result, standard test methods, fundamental analysis, design procedures, and quality evaluation are available to obtain acceptable results [1,2,3,4,5]. However, significant challenges still exist for the accurate quality assessment of compacted soil properties under in situ conditions.

Current in situ quality evaluation testing techniques include stiffness test methods [6,7,8,9,10,11,12,13]. Stiffness spot test methods are less frequently employed, as the number of test points is limited, certain test methods are time-consuming, and they cause interference with the construction operations. Intelligent compaction (IC) quality control that integrates a vibratory roller with an accelerometer-based measuring system, a high-precision global positioning system (GPS), and an on-board data acquisition system to monitor the compaction process is highly recommended [8,10]. IC quality control overcomes the limitations of traditional stiffness spot test methods. Studies suggest that stiffness spot test methods must be conducted to calibrate intelligent compaction measurement values (ICMVs). The calibration makes IC achieve 100% coverage for various subgrade soils utilized for construction, with benefits such as improved compaction, uniformity, reduced over/under compaction, and the ability to identify weak spots, as well as design, construction, and performance integration. However, stiffness spot test methods and IC do not detect the *w* and *C,* which significantly affect cement-stabilized subgrades [14,15,16].

Studies and experience show that *w*, *C,* and *ρ*_d_ can vary based on the construction method. The quality control process may sometimes be delayed because of anticipated problems, such as the organization of the crew to the site or faulty equipment [17,18]. The hydration reaction between *w* and *C* causes the stiffness, *w,* and *ρ*_d_ to change with *T.* As a result, it is essential to clarify the correct *C, w, ρ*_d_, and *T*. This finding ensures an accurate assessment of the stiffness properties to predict the improvement effects of the stabilized subgrade [3,19,20,21]. Below/above the optimum *w* and *C* requirements, the required engineering properties may not reach/exceed the design limit, which may produce different physical and stiffness characteristics of the stabilized subgrade [22]. It is purportedly reported that some of these characteristics might be unfavorable to certain dynamic and traffic-loading conditions, thus affecting the pavement life and performance [11,23,24,25,26,27,28]. Therefore, gaining insight into the relationship between these properties requires improved/new test methods that are capable of concurrently and instantaneously assessing the stiffness and physical properties.

Recently, integrated geophysical mechanical test methods have been employed to solve many geotechnical problems [29,30,31]. The most common integrated geophysical test method is soil resistivity (*ρ*). The *ρ* of compacted cement-stabilized soils depends on factors such as *w*, *ρ*_d_, porosity, and temperature [32,33]. The *ρ* of the cement-stabilized soil is affected by the changes in the compaction. *ρ* is selected to measure and characterize soil properties during in situ and laboratory tests [29,30,31]. Models are available to describe the *ρ* of several geomaterials, including cement soil mixtures [32,33,34]. Thus, *ρ* can be an alternative indicator of the changes in the mechanical and physical properties of cement-stabilized soil. Incorporated *ρ* cone penetrometer [29], triaxial [30,35], and consolidation devices [31] have been used to evaluate the mechanical and physical properties of cement-stabilized subgrade, which produced excellent results. However, there is a lack of equipment and methodological approaches for simultaneously and non-destructively assessing the *ρ*, soil stiffness, *w*, *C,* and *ρ*_d_ properties for in situ quality evaluation of cement-stabilized subgrade.

In this study, we aimed to develop and apply the resistivity plate loading test (*ρPLT*) to assess the properties of compacted cement subgrade. We further assessed the effect of variations in *ρ*_d_, *w*, *C,* and *T* on the stiffness of the cemented subgrade, adopting the Taguchi design of the experiment. Furthermore, mathematical relationships were developed to predict *K*_30_, *ρ*, the unconfined compression strength (*UCS*), and all the quality parameters by adopting grey rational analysis (*GRA*).

### Resistivity Plate Loading Device

Figure 1a shows a section of a typical resistivity plate loading device and Figure 1b shows a picture of the device, which comprises mild steel, a resistivity metre, and titanium electrodes (diameter of 4.0 mm and height of 7.0 mm) properly insulated with acetal plastic. The Wenner four-probe method of *ρ* measurement was employed. This technique is commonly applied in geology and soil science *ρ* measurements. Aside from the Wenner configuration method, several other electrode configuration methods are available in the literature and are rich in theory. Refs. [36,37] discuss the merits and demerits of the various electrode configuration methods. The suitability of a particular array depends on the signal-to-noise ratio, depth of investigation, geometry of the electrode, etc. The Wenner four-probe electrode arrangement was chosen, and instrumental sensitivity was not as important as in the other array geometries. The electrode spacing determines the depth of electrical current penetration and volume measurement. In a homogeneous soil, the soil volume measured is approximately πa^3^, where (a) is the electrode spacing. In an experiment, a resistivity meter is connected to the loading plate, i.e., two electrodes (A and B) for injecting current into the stabilized subgrade and two voltage electrodes (M and N) for measuring the potential difference. Equation (1) is used to convert the current and voltage to an apparent resistivity.
(1)$$\rho =K\frac{\Delta V}{I}$$
where *K* is the geometrical factor dependent on the electrode configuration; for the Wenner configuration, *K* = 2πa, π is 3.14, a is the electrode space, V is the voltage, and *I* is the current.

The *ρPLT* method can be used to assess a whole range of soils, including subgrade, subbase, base, and treated soils used for road and railway construction. A calibration procedure may be required to avoid errors in *ρ* measurements when *ρPLT* is conducted on soils in rocks [38]. The electrodes of the resistivity plate loading device may also suffer contact resistance problems when installed on frost, very dry soils, and soils in rock [38,39].

In an experimental setup, *ρ* measurement errors have two origins: (1) poor electrode contact and (2) instrumental noise. The electrode depth (7.00 mm) and diameter (4.00 mm) induce minor faults in the former. In the latter, measurement time causes minimal or no variation in the *ρ* measurement. The properties of the electrode geometry and configuration are similar to other laboratory measuring instruments available in the literature [40].

Acetal plastic has excellent mechanical and low moisture absorption properties. Additionally, the coefficient of friction of acetal is within the range of metals that are commonly employed for static loading plates. As a result, the acetal plastic may have minimal or no effect on the stiffness measured with this device.

## 2. Experimental Program, Materials, Soil Preparation, and Testing

### 2.1. Experimental Program

The experimental program involves (1) testing the basic properties of the soil, (2) determining the most significant factors that affect the compaction quality of the cement-stabilized subgrade and their factor levels, and (3) selecting an orthogonal array and running the experiments based on the orthogonal array. The experiments conducted included the unconfined compression strength (*UCS)* test and resistivity subgrade reaction modulus test (*ρK*_30_).

### 2.2. Soil Properties

The soil used for this study was obtained from Hebei, China. The soil is classified as A-2-4 according to the American Association of State Highway and Transportation Official (ASSHTO) and SP-SM according to the Unified Soil Classification System (USCS). Figure 2 shows the particle size distribution curve. The liquid limit is 29.52% and the plastic limit is 19.19%. The uniformity coefficient (Cu) is 48.46l, the coefficient of gradation (Cc) is 0.57, and the group index (GI) is 0.

ASTM D558-96 procedures were followed to determine compaction characteristics. The test results were applied to determine the maximum dry density (MDD) and optimum moisture content (OMC). The test was conducted by adding ordinary Portland cement of grade 32.5 to the soil in proportions of 5%, 8%, and 12% by weight of the total soil dry mass. The mixture was thoroughly mixed until uniformity was achieved. Figure 3 shows the compaction characteristics. Twelve percent *C* was considered to give the best results. Ref. [37] recommended adding 5–12% *C* to soils with similar properties.

### 2.3. Development of Taguchi Orthogonal Array

The *ρ*_d_, *C*, *w*, *T,* and other factors, such as the particle size and shape of the soil, type of cement, porosity, temperature, etc., affect the in situ strength and compaction quality control of the cement-stabilized subgrade [17,18,41]. The Taguchi method may be employed to assess the essential factors for compaction quality control and quality assurance. The Taguchi method works on the principles of an orthogonal array and provides fewer variances in results, with an optimal set of control parameters. The principle allows the characterization of complex behavior, maximizing the test coverage while minimizing the test cases to be considered and the ease of data analysis [42,43]. Table 1 presents the factor levels, and Table 2 presents details of the Taguchi orthogonal array design used in this study. The control factors were carefully selected based on previous research [1,44,45]. *ρ*_d_, *w*, *C,* and *T* were selected as control input parameters, and their corresponding levels were determined as shown Table 1.

The Taguchi method works best in processes and procedures where one quality characteristic is to be evaluated. However, many practical applications are a compendium of quality characteristics. Multicriteria decision-making methods are often utilized in such complex scenarios. Recommended multicriteria decision-making methods include grey relational analysis (*GRA*), techniques for order preferences by similarity to ideal solution (TOPSIS), the analytic hierarchy process (AHP), etc. More than one quality characteristic was assessed, turning the multiple responses into a single response problem. Multicriteria decision-making methods have been suggested to be useful, easy, and efficient when integrated with the Taguchi method [46], with the most common methodologies being *GRA* and TOPSIS with several modifications. The *GRA* method was employed in this study to combine all the multiple response values into a single response value; thus, an interrelationship between the responses based on the grey relational coefficient was obtained. The *GRA* method has the following advantages: the results depend on the original values of the measured responses; the calculations are simple and suitable for multiple complicated relationships between responses and fairness comparison; and dimensional attributes are ensured by normalization. The steps to employ the *GRA* technique are as follows:

Step 1. Define a set *A* of n alternatives concerning *W* evaluation criteria as follows
(2)$$A={\left[b_{ij}\right]}_{m+n}=\left[\begin{array}{cccc} b_{11} & b_{12} & \cdots & b_{1m}\\ b_{12} & b_{22} & \cdots & b_{2m}\\ \vdots & \vdots & \ddots & \vdots \\ b_{n1} & b_{n2} & \cdots & b_{nm} \end{array}\right](i=1,2,\ldots ,n\;and\;j=1,2,\ldots ,m)$$
where $b_{ij}$ represents the response value of the ith alternative on the jth criterion.

Step 2. If the quality characteristic expectancy maximizes the response, use the larger, the better Equation (3). Use the smaller, the better Equation (4) when the quality characteristic minimizes the response. Use Equation (5) for nominal quality characteristics.
(3)$$\bar{b_{ij}}=\frac{b_{ij}-\min(b_{ij},i=1,2,\ldots ,n)}{\max\left(b_{ij},i=1,2,\ldots ,n\right)-\min\left(b_{ij},i=1,2,\ldots ,n\right)}$$
(4)$$\bar{b_{ij}}=\frac{Max(b_{ij},i=1,2,\ldots ,n)}{\max\left(b_{ij},i=1,2,\ldots ,n\right)-\min\left(b_{ij},i=1,2,\ldots ,n\right)}$$
(5)$$\bar{b_{ij}}=\frac{\left(\left|b_{ij}-T\right|\right)-\min\left(\left|b_{ij}-T\right|\right),i=1,2,3,\ldots ,n}{\max\left(\left|b_{ij}-T\right|,i=1,2,\ldots ,n\right)-\min\left(\left|b_{ij}-T\right|,i=1,2,\ldots ,n\right)}$$

Step 3. Compute the grey relational coefficient (*GC*) for normalization using Equation (6).
(6)$$GC_{ij}=\frac{\Delta_{\min}+\lambda \Delta_{\max}}{\Delta_{ij}+\lambda \Delta_{\max}}$$

Step 4. Compute the grey rational grade ($G_{i}$) using Equation (7).
(7)$$G_{i}=\frac{1}{m}\sum GC_{ij}$$
where m is the number response; $\Delta_{\min}$ is minimum value of Δ; $\Delta_{\max}$ is maximum value of Δ; λ is the distinguishing coefficient which is defined in the range 0≤λ≤1, (in this study 0.5 was used); $GC_{ij}$ is the gray relational coefficient for the ith experiment and jth response; *T* is the target value. In this study, average response was used.

### 2.4. Soil Preparation

Large soil lumps were broken with a wooden mallet, oven-dried at a temperature of 105 °C, and allowed to cool. All calculations were based on the oven-dried soil mass. The required amount of soil and *w* were calculated, thoroughly mixed in a bowl, transferred into plastic bags, and then tightly sealed. The samples were then kept for a day for moisture equilibration. Afterwards, the required amount of *C* was calculated, added, and mechanically mixed until the clumps formed were less than 5% of the total mix.

Samples prepared for the model test were cast into the modified soil consolidation soil chamber unit and subjected to an impact compaction test. The weight of the material in the container was calculated. The mixture was compacted in three equal layers. In each layer, a temperature sensor that was properly insulated was installed to monitor the temperature evolution during the curing process. The temperature evolution has an important consequence on the strength development and *ρ* measurement [47,48]. The specimens were cured in an open dry place under sealed conditions. All samples were cast into a thin, insulated steel cylinder, and their upper surfaces were tightly sealed using a plastic cover.

Two (2) samples were prepared for the UCS test, adopting the sample plan in Table 2. The ASTM D 1633 (2014) standard requires samples with a diameter, height, and height-to-diameter ratio (h/d) of 101.60 mm, 116.40 mm, and 1.15, respectively, or samples with a diameter, height, and h/d ratio of 71.10, 142.20 mm, and 2.00, respectively. In this study, samples were prepared at a height of 80.00 mm and diameter of 39.10 mm with an h/d ratio of 2.05. The height and weight of the prepared samples were within an accuracy of ±1%. The prepared samples were stored in a zip-lock polythene bag under damp conditions. The mixing, compaction process, and each test was completed in less than two (2) hours. The UCS was tested in a triaxial loading frame with a computerized data-acquisition system at a loading rate of 1.00 mm/min (ASTM D1633).

### 2.5. Test Setup

Figure 4c shows the details of the test setup, which comprises a loading system (air cylinder, air compressor, air pressure regulator, and air pressure gauge), deformation measurement system (linear variable displacement transducers (LVDTs)), computerized data-acquisition system, *ρPLT* device, temperature sensors (embedded within the compacted soil; bottom, middle, and upper-lower with rubber on it), and a soil-containing unit (diameter of 25.20 cm; height of 27.00 cm).

The temperature sensors utilized a Pro Led DC 12 V digital thermometer and sensor probes with a temperature range of −50 to 110 °C.

The soil-containing unit was constructed of a steel frame with a height of 30.0 cm and a diameter of 25.2 cm. A plastic bag was inserted into the container, and the soil was packed into and compacted using the impact method. The plastic bag prevented the walls from conducting electrical current during testing. The walls of the soil container were also such that they exerted minimal or no effects on the stresses and strains during testing.

The soil-containing unit edges exert a border effect on the infinite value of the resistivity of air. These border effects are significant challenges for laboratory *ρ* measurements. Solutions have been developed in [49]. The *ρ* measured in our study is affected by this border effect. However, we do not intend to correct this measurement. The testing conditions are similar to those likely to be experienced for in situ conditions in a typical railway or a road embankment where the geometry may be a 2- or 3-dimensional structure surrounded by air. Accounting for these possible conditions will be complicated.

## 3. Testing Process

### 3.1. Temperature

The temperature sensors were monitored every fifteen minutes for the first two hours. Afterwards, the temperature sensors were read every two hours during the daytime until the curing periods ended.

### 3.2. Resistivity Subgrade Reaction Modulus Test

*ρPLT* was selected to assess the properties of the model samples. The subgrade reaction modulus (*K*_30_) test was conducted according to the procedures of TB10621-2014. The required load was applied through the piston to the load plate from an air cylinder supported by a reaction frame. The air cylinder was regulated through the air regulator. When the required air was supplied to the air cylinder, the valve that opened to the air cylinder was closed until the next load application was opened. This step was repeated throughout the test. LVDTs with an accuracy of 0.01% of the full range (100 mm) were mounted on three (3) sides of the load plate and configured to a computer that automatically displayed the deformation readings. The average deformation reading was employed for all calculations. For accuracy, all the devices were calibrated after three (3) sample runs during testing. The equivalent *K*_30_ was calculated with Equation (8).
(8)$$K_{30}=\frac{\sigma }{s_{D}}$$
where *K*_30_ is the equivalent subgrade reaction modulus, *s*_D_ = 1.25 mm multiplied by (D/30 cm); D is the diameter of the resistivity loading plate, and *σ* is the stress at *s*_D_.

*ρ* was measured with an ETCR3000B digital grounding resistance soil resistivity tester. *ρ* was recorded five minutes after every load application, which reduces the variability of *ρ*.

### 3.3. Unconfined Compression Strength Test

After curing the samples for 28 days, the samples were immersed in water for no less than four (4) h. Afterwards, the soil samples were removed from the water, dampened, and tested. During testing, loads were applied until the load at failure was recorded. The UCS strength was calculated using Equation (9). The ASTMD 1633 recommends multiplying the samples prepared with method B by 1.10 to obtain the calculated UCS strength. Based on laboratory studies, [50] suggested multiplying samples with an h/d ratio of 2.0 by a factor of 0.86. This step was purposely applied to convert the strength of an h/d ratio of 2.00 to that of the h/d ratio of 1.15, which is commonly utilized in routine soil-cement testing. In this study, a factor of 1.10 was applied.
(9)$$q_{u}=\frac{P\left(1-\varepsilon \right)}{A_{0}}\times F$$
where *q_u_* is the UCS strength; P is the force applied; *ε* is axial strain; *A*_0_ is the cross-sectional area of the specimen; and *F* the strength correction factor.

### 3.4. Explanation of Test Results

#### 3.4.1. Compaction

The compaction characteristics of soils are evaluated regarding OMC and MDD. Figure 3 shows the compaction characteristics. The sample with 0% cement content exhibited the highest MDD compared with the other samples. The relatively low MDD of the samples with cement additives, particularly 12%, can be attributed to (1) the cement additive, which causes aggregation of the soil particles to occupy large spaces that change the effective particle size of the soil; (2) the absorption of the compaction energy by the hydration products of the sample (the OMC increases with the addition of *C* due to the increase in *w* required for cement hydration); and (3) the loss of moisture due to evaporation from heat generated during hydration [13,51].

#### 3.4.2. Temperature

Figure 5 shows the temperature–time plot history for the tested samples. The sample with the highest temperature exhibited a peak temperature of 13 °C (1) and a low temperature of 1.0 °C (3). An exothermic chemical reaction between cement and water occurs when they are blended. Temperature, relative humidity, type of cement, and *C* can affect the reaction. The relatively low temperature observed was caused by environmental factors [52]. High-temperature conditions enhance cement hydration and pozzolanic reactions in cement-stabilized soils, thus improving the strength properties. However, very high curing temperatures (20–60 °C) have caused cement hydration products to be arranged randomly, thus producing large pores in soils (crossover effect) [48]. Elsewhere, lower temperature conditions have also been reported to contribute to poor strength development [53]. The different temperature gradients affect the strength development and *ρ* measurement [48].

#### 3.4.3. Resistivity Subgrade Reaction Modulus

The study clearly showed the dependence of stiffness and *ρ* on *ρ*_d_, *w*, *C,* and *T*. All *ρ* readings were temperature corrected using Equation (10). Figure 6a presents a typical stress- and temperature-corrected *ρ*, and the change in *ρ* with successive load application was a kind of multistage process during testing. We envisaged that the contacts between the electrodes and the soil would improve with successive loads acting on the loading plate. As forces were applied to the loading plate, comparable *ρ* changes were observed. The changes in *ρ* were analogous to the changes in the soil properties, interaction of the *w*, cement hydration products, and influence of the applied stress beneath the loading plate. For a given *T*, the higher the cement content is, the greater the number of hydration compounds, thus the higher the observed *ρ* [32]. Additionally, the lower *w* is, the higher the observed *ρ*. The selected *ρ* was calculated by establishing a linear equation between the measured *ρ* and the deformation. *ρ* is determined at the equivalent deformation used to calculate the subgrade reaction modulus. The test results are summarized in Table 2.
(10)$$\rho_{w}\left(t\right)=\frac{\rho \left(t_{0}\right)}{1+\alpha \left(t-t_{0}\right)}$$
where α≈0.025 (°C^−1^), $\rho_{w}\left(t\right)$ is the temperature-compensated electrical resistance, $t_{0}$ is a fixed reference temperature, and $\rho \left(t_{0}\right)$ is the *ρ* at a temperature $t_{0}$.

Figure 6a,b show the stress and resistivity curves and stress–strain curves for the tested samples. The samples exhibited different forms of stress and strain characteristics. Generally, strain hardening was observed for samples compacted at a higher *w* and low *C*. A similar observation was observed for certain samples with a low *w*. A linear elastic behavior was also observed for certain samples compacted at a higher *ρ*_d_.

#### 3.4.4. Unconfined Compression Strength Test

The UCS is an index for quantifying the effectiveness of an additive [2,54] on the soil strength. The sample must not collapse, or no significant loss in strength should be observed during water immersion. The test results for UCS are presented in Table 2. The results range from 0.85 to 8.18. The highest results were associated with samples compacted at the highest *ρ*_d_. The *w* and *C* content variation had a different effect on the UCS. UCS values in the range of 0.2–0.4 MPa are usually permitted for subgrade applications. However, considering the different compositions of the samples, they may have negative consequences on the performance of the compacted subgrade.

## 4. Discussion

### 4.1. Range Analysis

Range analysis was performed in the Minitab statistical package to assess the influence of the various factors on the response variables and their ranks, as shown in Table 3. The rank based on the Delta statistics compared the relative magnitudes of their effects. The delta statistic is the highest average minus the lowest average for each factor. The ranks and average level responses are used to determine which element provides the best results. All the means of the mean plots were determined using the nominal characteristics. Furthermore, all ANOVAs were performed at a confidence level of 95% and 5% significance level.

#### 4.1.1. Subgrade Reaction Modulus

In Table 3, *ρ*_d_, *T*, *w*, and *C* are ranked in order of importance. The analysis of variance for the individual factor contributions and their F values of 1.74, 1.16, 0.67, and 0.62 are presented in the order of the ranks. As shown in Figure 7, a combination of 23 *ρ*_d_, 5% *w*, 18% *C*, and *T* of forty-eight (48) h is considered to give the best result. According to the literature, the strength of the cement-stabilized subgrade is related to the solid phase, *w*, *C,* and *T* [19,55]. An increase in *ρ*_d_ yielded an increase in *K*_30_, as shown in Figure 7. An increased *ρ*_d_ caused interlocking between the soil particles and cement particles, thus increasing the ability of the soil to resist the mechanical forces that act on it [56,57]. Additionally, an increase in *w* weakens the cohesion strength between the particles of the soil and the hydration products [18].

#### 4.1.2. Resistivity

Table 3 shows the ranks of the input factors. The analysis of variance on the individual factor contributions and their F values according to the ranks (*C*, *T*, *w,* and *ρ*_d_) are 3.19, 0.94, 0.46, and 0.38, respectively. *C* exhibits a significant effect on the *ρ* response. As shown in Figure 8, the line slope shows that the effect and the degree of the slope are comparative to the magnitude of the inputs. *ρ* increases with an increase in *ρ*_d_ and *T* and decreases with an increase in *w*. Several others have reported the same findings [32]. The *C* produces an amount of hydration products for a given *T,* resulting in a denser structure. The higher *T* is, the higher the hydration products produced. Thus, the free water space and porosity decrease and tortuosity increases, increasing *ρ*. Additionally, an increase in *ρ*_d_ causes interlocking and parking of the soil particles, which renders the electrical flow path tortuous, increasing *ρ*.

#### 4.1.3. Unconfined Compressive Strength

Since all the samples were assessed at 28 days, the effect of curing was excluded. Subsequently, *ρ*_d_, *w*, and *C* were evaluated to determine their impact on the UCS strength development. Table 3 shows the ranks of the input factors. The F values of the input factors according to the ranks are 5.93, 0.54, and 0.23. Figure 9 presents the main effect plots. The best strength was achieved at 1.80 *ρ*_d_, 18% *w,* and 12% *C*. Several researchers using the UCS test have investigated the strength of cement-stabilized soil [44,58,59,60]. Almost all researchers reported an increase in strength with an increase in *ρ*_d_, and our results confirm this finding. The UCS strength increases with an increase in *ρ*_d_ and *w* up to 12% *C*, as shown in Figure 9. At 18% *C*, the UCS decreases, which is contrary to the results presented in [44,53,58,59,60,61,62]. This observation can be explained by the observation that at higher C, the cement-treated soil requires enough *w* to undergo complete hydration [61,63]. Furthermore, at higher C, microcracks develop, easily propagating and joining under applied loads and thus reducing the UCS. This finding means that there is an optimum amount of *w* and *C* for which the number of bonds and their configuration offer the best strength development. Any deviation from these values will negatively impact the strength development.

### 4.2. Grey Rational Analysis

Once calculations for GC and G were finished, their values were entered in Table 4. ANOVA was performed to assess the effects of the experimental input parameters on the GRA grade. The results of the ANOVA are presented in Table 5. *ρ*_d_, *T*, *w,* and *C* influenced the GRA grade values with contributions of 79.09%, 10.30%, 3.94%, and 6.67%, respectively. A GRA near 1 in Table 4 can be considered a criterion that gives the best optimal conditions for *K*_30_, *ρ,* and *UCS*.

### 4.3. Regression Analysis

In this study, regression analyses were employed for modelling and predicting the response variables. Two different models were initially proposed: linear interactions and linear plus interactions. The best fitting models were selected. The highest performing predictive equations obtained from the analysis are given below. The equation reveals the nature of the relationships of the input parameters and their effect on the responses. Directly proportional responses are acknowledged using a + sign, while the - sign represents the opposite. Equations (11)–(14) were identified as the best-performing equations for *K*_30_, *ρ*, *UCS*, and *GRA*, respectively. The equations predicted the average *K*_30_, *ρ*, *UCS*, and *GRA* with adjusted coefficients of correlation (R^2^) of 72.18%, 78.09%, 71.06%, and 55.63%, respectively. The individual unique predictor contributions for the equations are shown in Table 6.
(11)$$K_{30}=133.86+20.21\rho_{d}+8.32T-20.20\rho_{d}w+2.0wC-14.87w$$
*Ρ* = 81.40 + 12.87*ρ*_d_ + 13.30*T* − 4.0*ρ*_d_*w* + 20.23*C* − 11.30*w*(12)
(13)$$UCS=3.34+1.21\rho_{d}-0.95wC+0.54w$$
(14)$$GRA=0.5749-0.203\rho_{d}+0.0004wC+0.0238wT-0.0586CT+0.0418\rho_{d}w$$

### 4.4. Implications for Practice

Several studies have suggested that *K*_30_ can reflect the strength, stiffness, and performance of compacted subgrade in China. However, we suggest that *K*_30_ is the main parameter that is commonly employed but is not the only parameter that can assure compacted subgrade stiffness, performance, and deformation characteristics. Compacted soil typically remains in the elastic stage, away from the failure stage, as evidenced by the plate loading tests (refer to Figure 6b). As a result, using this indicator alone to capture the strength and stiffness characteristics is deemed ineffective. Additionally, the *K*_30_ test results are affected by factors such as *w*, *ρ*_d_, *T*, and *C* (refer to Figure 7) for the cement-stabilized subgrade. The variability of *w*, *ρ*_d_, and *C* is reported to cause subgrade problems, such as swelling, subgrade cavity, and differential settlement.

There is no good indicator to reflect the changes in the physical and stiffness properties of the subgrade, particularly the *K*_30_, w, *T*, *C,* and *ρ*_d_ of the cement-stabilized subgrade for construction. However, practitioners are aware of how *w*, *T, C,* and *ρ*_d_ affect the test outcome of *K*_30_ and the performance of the compacted subgrade. Nonetheless, a qualitative method to instantaneously and concurrently assess the *w*, *T*, *C,* and *ρ*_d_ effects with *K*_30_ has always been associated with certain problems. Our test method solves this problem and serves as an advancement in compaction measurement systems and in situ testing technologies.

The proposed regression Equations (11)–(13) are presented to predict the average responses of *K*_30_, *ρ,* and UCS, which showed high performance with R^2^ values of 72.18%, 78.09%, and 71.06%, respectively. Use of the proposed equations is encouraged during the construction of cement-stabilized subgrade for similar soils utilized in this study with similar *w* and *C* requirements. Equation (14) was introduced as a unique equation that can give a combined unique identity to *K*_30_, UCS, and *ρ*, considering the input factors employed in the study with an R^2^ of 55.63%. The test results presented in this study conclude that the proposed remediation was significant. Although our test method may be slow, its benefits outweigh those of commonly known methods in the literature.

## 5. Conclusions

A resistivity plate load device was developed to evaluate the properties of the cement-stabilized subgrade. Experimental studies were conducted on compacted cement soils according to the Taguchi orthogonal L9 array. Multivariate stepwise regression models were developed to predict *K*_30_, *ρ,* and UCS considering *ρ*_d_, *T*, *w,* and *C* for compaction quality evaluation purposes. Our conclusions are as follows.

(1)The resistivity plate load device can be efficiently used to assess the *w*, *C*, *ρ*_d_, and *T* effects on *K*_30_ for the cement-stabilized subgrade for construction. Therefore, the device is recommended for field use. This recommendation must be evaluated and validated in the field with a vast variability in compacted soils with variable *w*, *C*, and *ρ*_d_ properties that may confirm the implications of our laboratory observation.(2)*K*_30_ is significantly affected by *C*, *w*, *ρ*_d_, and *T*. It is important within subgrade construction control protocols to confirm that compacted cement-stabilized subgrade C, *ρ*_d_, *T*, and *w* are considered for a well-informed decision on *K_3_*_0._(3)Different values of *C*, *ρ*_d_, and *w* have different effects on the USC, leading to variable USC strength values and confirming the need to control *C*, *ρ*_d_, and *w* during construction for enhanced performance.(4)The proposed regression models showed better performance, and therefore, are recommended to predict the average *K*_30_, *ρ*, UCS, and GRA grades during construction of the cement-stabilized subgrade for similar soils.

The statistical analysis presented in this study produced excellent results. The analysis should be treated as a simplified approach and a general guide that can be adopted for field compaction quality control for the cement-stabilized subgrade. Future studies include a field application of the test method and calibration of ICMVs for compaction management of the cement-stabilized subgrade. Notwithstanding, this study was a significant step towards providing a reliable reference for engineering practice in compaction quality control and a considerable step towards advancement in compaction measurement systems and in situ testing technologies.

## Figures and Tables

**Figure 1 materials-15-03453-f001:**
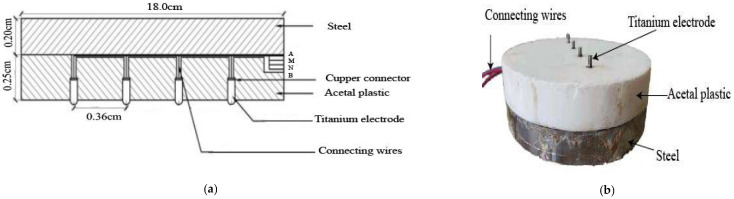
Resistivity plate loading device. (**a**) A section of the laboratory resistivity loading plate and (**b**) a picture of the resistivity loading plate.

**Figure 2 materials-15-03453-f002:**
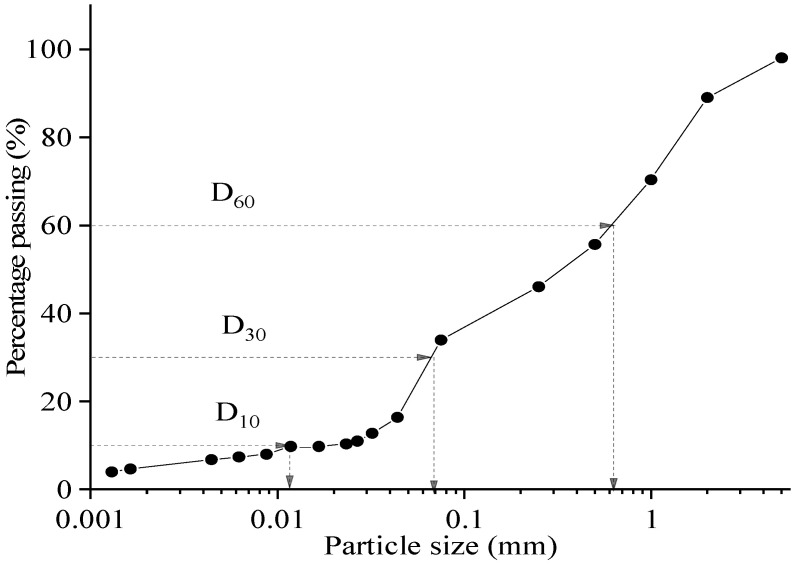
Particle size distribution curve.

**Figure 3 materials-15-03453-f003:**
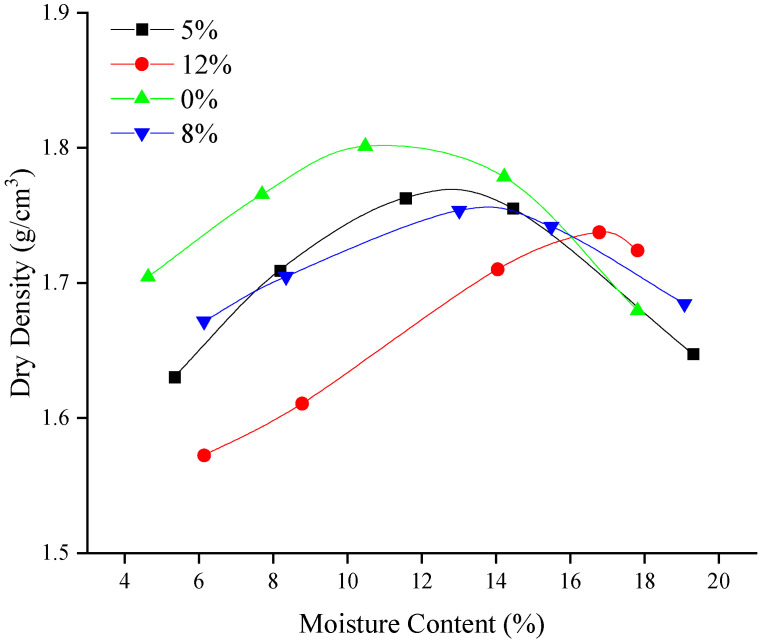
Compaction characteristic of the compacted soils.

**Figure 4 materials-15-03453-f004:**
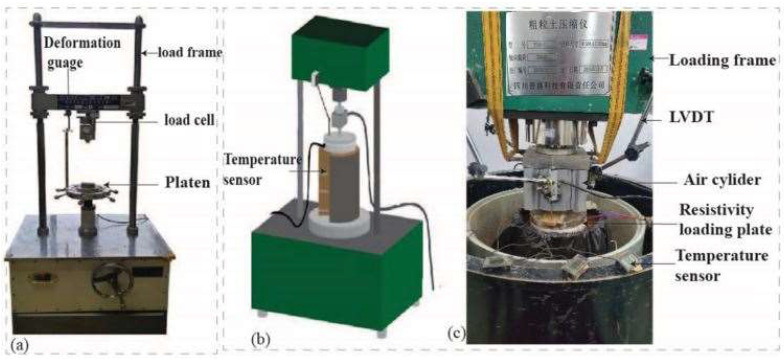
Experimental device. (**a**) UCS testing machine; (**b**) 3D drawing of the model test; (**c**) picture of the model test.

**Figure 5 materials-15-03453-f005:**
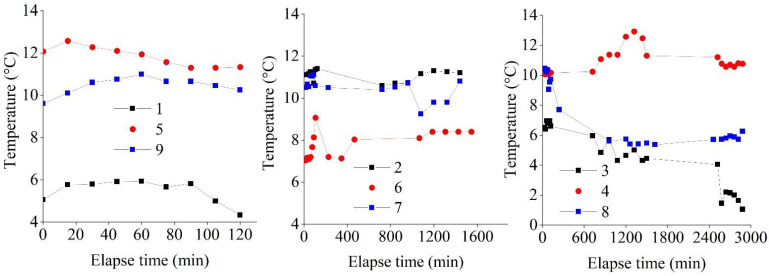
Temperature–time history plots for the tested samples.

**Figure 6 materials-15-03453-f006:**
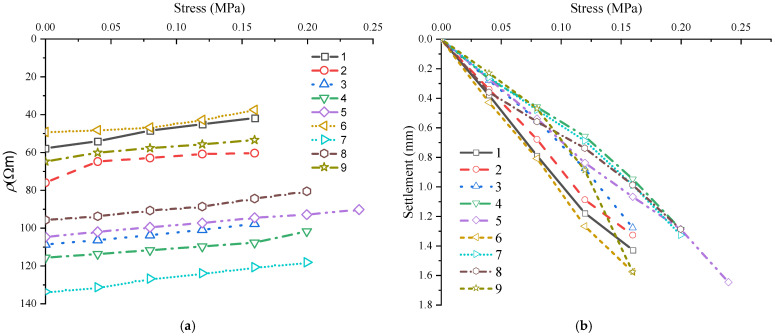
Resistivity plate loading test results (**a**) typical of stress- and temperature-corrected apparent resistivity curves from the concurrent resistivity subgrade reaction modulus test and (**b**) typical of the stress strain curves.

**Figure 7 materials-15-03453-f007:**
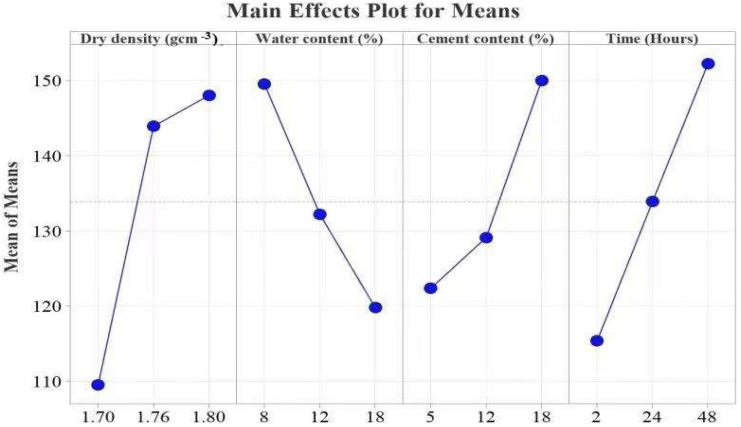
Effects of the control factors on the *K*_30_.

**Figure 8 materials-15-03453-f008:**
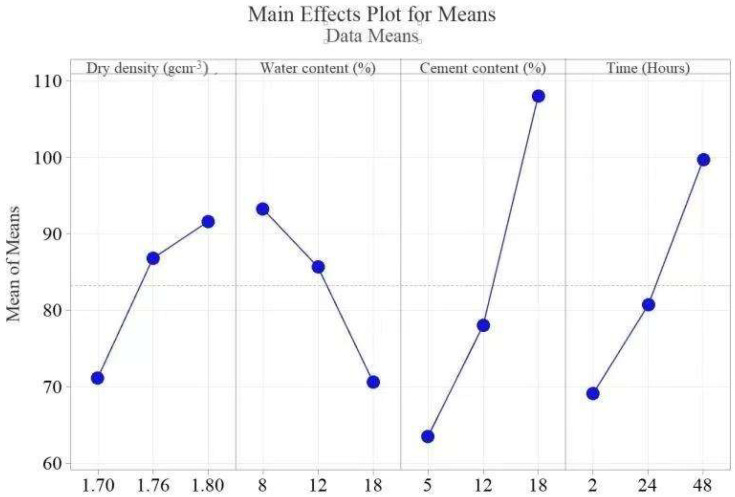
Effects of the control factors on the *ρ*.

**Figure 9 materials-15-03453-f009:**
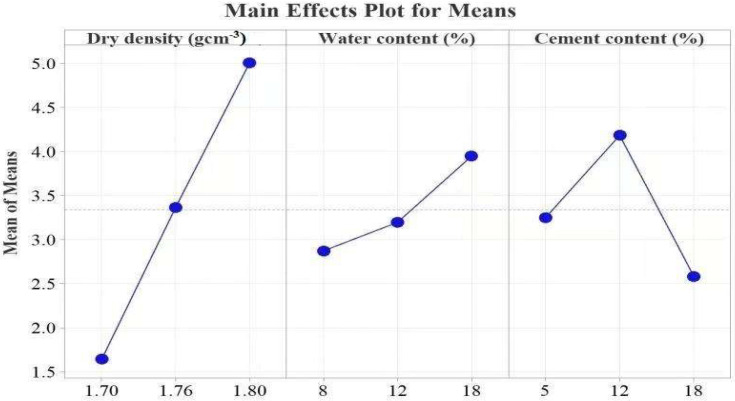
Effects of the control factors on USC.

**Table 1 materials-15-03453-t001:** Experimental factors and their levels.

Factor	Level 1	Level 2	Level 3
*ρ* _d_	1.70 g cm^−3^	1.76 g cm^−3^	1.80 g cm^−3^
*w*	8%	12%	18%
*C*	5%	12%	18%
*T*	2	24	48

**Table 2 materials-15-03453-t002:** Taguchi L9 Orthogonal array for conducting the design of experiments and test results.

ID	Taguchi Orthogonal Array				Test Results					
	*ρ*_d_ (g·cm^−3^)	*w* (%)	*C* (%)	*T* (h)	K30 (MPa/m^2^)		*ρ* (Ωm)		UCS (MPa)	
					Average	St.d	Average	St.d	Average	St.d
1	1.70	5.0	8.0	2.0	95.33	0.67	47.10	1.43	0.88	0.03
2	1.70	12.0	12.0	24.0	103.3	3.33	66.29	0.77	3.12	0.06
3	1.70	18.0	18.0	48.0	130.10	2.67	100.00	2.89	0.93	0.04
4	1.76	5.0	12.0	48.0	173.33	0.00	108.20	0.62	3.18	0.04
5	1.76	12.0	18.0	2.0	140.00	1.67	100.00	0.11	2.26	0.08
6	1.76	18.0	8.0	24.0	118.50	1.50	52.10	2.18	4.66	0.10
7	1.80	5.0	18.0	24.0	179.98	2.68	123.99	0.09	5.66	0.13
8	1.80	12.0	8.0	48.0	153.34	2.66	90.94	2.21	4.21	0.06
9	1.80	18.0	12.0	2.0	110.83	2.50	59.27	0.26	8.25	0.08

St.d is standard deviation.

**Table 3 materials-15-03453-t003:** Response table for mean.

Subgrade Reaction Modulus (*K*_30_)				
Level	*ρ* _d_	*w*	*C*	*T*
1	109.60	149.50	122.40	115.40
2	143.90	132.20	129.20	133.90
3	148.00	119.80	150.00	152.20
Delta	38.50	29.70	27.60	36.90
Rank	1.00	3.00	4.00	2.00
Soil electrical resistivity (*ρ*)				
1	71.27	93.25	63.52	69.12
2	86.78	80.31	72.69	75.36
3	91.59	70.64	108.00	99.72
Delta	20.32	22.60	44.48	30.61
Rank	4	3	1	2
Unconfined compression strength test (USC)				
1	1.64	2.87	3.25	
2	3.37	3.20	4.18	
3	5.01	3.95	2.58	
Delta	3.36	1.07	1.60	
Rank	1.00	3.00	2.00	
GRA				
1	0.82	0.55	0.57	0.61
2	0.52	0.67	0.64	0.62
3	0.42	0.53	0.54	0.52
Delta	0.40	0.14	0.09	0.11
Rank	1.00	2.00	4.00	3.00

**Table 4 materials-15-03453-t004:** Grey rational analysis.

ID	*K* _30_	*ρ*	UCS	*K* _30_	*ρ*	UCS	GRA G
	Normalized			Gray Relational Coefficient			
1	0.18	0.16	0.05	0.74	0.76	0.90	0.80
2	0.00	0.00	0.00	1.00	1.00	1.00	1.00
3	0.24	0.05	0.77	0.67	0.92	0.39	0.66
4	0.96	0.85	0.23	0.34	0.37	0.68	0.46
5	0.77	0.77	0.00	0.39	0.39	0.99	0.59
6	0.64	0.31	0.60	0.44	0.61	0.45	0.50
7	1.00	1.00	0.58	0.33	0.33	0.46	0.38
8	0.85	0.68	0.49	0.37	0.42	0.50	0.43
9	0.60	0.38	1.00	0.46	0.57	0.33	0.45

**Table 5 materials-15-03453-t005:** ANOVA Analysis of GRA grade.

Factor	DF	Adss	Ad MS	F α = 0.5	*p* Value	% Contribution
*ρ* _d_	2	0.261	0.131	11.22	0.009	79.09
*w*	2	0.034	0.017	0.35	0.717	10.30
*C*	2	0.013	0.007	0.03	0.883	3.94
*T*	2	0.022	0.011	0.21	0.817	6.67
Error	-	-	-			
Total	8	0.33				100.00

**Table 6 materials-15-03453-t006:** Variance analysis of the regression equations.

Subgrade Reaction Modulus (*K*_30_)						
Source	DF	Adj SS	Adj MS	F-Value	*p*-Value	% Contribution
Regression	5	6538.43	1307.69	5.15	0.104	89.57
*ρ* _d_	1	1429.90	1429.90	5.63	0.098	19.59
*T*	1	242.29	242.29	0.95	0.401	3.32
*ρ* _d_ *w*	1	859.12	859.12	3.38	0.163	11.77
*wC*	1	8.03	8.03	0.03	0.870	0.11
*w*	1	1326.70	1326.70	5.23	0.106	18.17
Error	3	761.64	253.88			10.43
Total	8	7300.07				100.00
Soil electrical resistivity (*ρ*)						
Regression	5	6149.24	1229.85	5.90	0.087	90.77
*ρ* _d_	1	994.34	994.34	4.77	0.117	14.68
*T*	1	424.27	424.27	2.04	0.249	6.26
*ρ* _d_ *w*	1	16.12	16.12	0.08	0.799	0.24
*C*	1	982.29	982.29	4.71	0.118	14.50
*w*	1	766.37	766.37	3.68	0.151	11.31
Error	3	625.02	208.34			9.23
Total	8	6774.26				100.00
Unconfined compressive strength (*UCS*)						
Regression	3	20.93	6.9778	7.55	0.03	81.89
*ρ* _d_	1	5.47	5.4783	5.93	0.06	21.40
*wC*	1	2.24	2.2373	2.42	0.18	8.76
*w*	1	1.73	1.7281	1.87	0.23	6.77
Error	5	4.62	0.9245			18.08
Total	8	25.56				100.00
Grey rational analysis (GRA)						
Regression	5	0.225396	0.045079	3.01	0.197	83.36
*ρ* _d_	1	0.108580	0.108580	7.24	0.074	40.16
*wC*	1	0.001064	0.001064	0.07	0.807	0.39
*wT*	1	0.000019	0.000019	0.00	0.974	0.01
*CT*	1	0.008024	0.008024	0.54	0.517	2.97
*T*	1	0.009048	0.009048	0.60	0.494	3.35
Error	3	0.044987	0.014996			16.64
Total	8	0.270383				100.00

## Data Availability

Data are available from the authors upon request.

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
