# Peer review of "A Resistivity Plate Loading Device for Assessing the Factors Affecting the Stiffness of a Cement-Stabilized Subgrade"

_materials, 2022, doi:10.3390/ma15103453_

Round 1

Reviewer 1 Report

The authors of the article describe an interesting issue concerning the assessment of factors influencing the stiffness of the substrate stabilized with cement. As a result, they developed a device for evaluating the soil stabilization properties. A very important issue, well discussed, however, it is possible that, due to the considerable volume of the work, some errors appeared. They should be corrected and the article will meet the criteria.

Reviewer's comments

There is a citation error on line 32, page 2, double entry 21

Figure 1. The description should start with a capital letter

Page 5 line 18, citation, same number twice

Figure 3, figure 8, figure 9 The description should be uppercase

DOI numbers and links are missing in the literature

Reviewer 2 Report

It is a very poor paper: the structure does not comply with standards for scientific papers. The abstract is too long and not clear. The Introduction does not cite interesting and relevant works, moreover some cited references are not discussed.

Some acronyms appear but they are not defined. 

Figure 1 seems not to be in scale. Several messages "Error! Reference source not found." appear in the manuscript. I know the mean of "soil is classified as A-2-4", but this sentence shoyuld be explained and referenced.

What does "desing" mean in Figure 5?

The conclusion section is not well written and structured.

The English language should be deeply revised.

In my opinion the paper should be rejected.

Reviewer 3 Report

See the attached PDF comments.

Major revision & a re-review are needed.

Reviewer 4 Report

This article is very interesting, and the authors adequately explain the process of the new method developed. In my opinion, a photograph should be added to Figure 1 to properly understand the laboratory equipment.

In my opinion, the conclusions section is very reduced and the authors could expand each of the specific conclusions that they have included

Round 2

Reviewer 2 Report

In my opinion the paper should be rejected.

Author Response

Please,  the article has undergone English editing and the revised version is attached including the English editing certificate.

Reviewer 3 Report

Accept - The authors have addressed this Reviewer's comments.

Author Response

Please, the article has under English language editing, and the revised version is attached. Also, an English editing certificate has already been submitted to the handling Editor.
